# Understanding Cellular Respiration through Simulation Using Lego® as a Concrete Dynamic Model

**Michiel Dam [1,\*], Koen Ottenhof [1], Carla Van Boxtel [2] and Fred Janssen [1]**

[1] ICLON Graduate School of Teaching, Leiden University, Kolffpad 1, 2333 BN Leiden, The Netherlands; k.w.ottenhof@iclon.leidenuniv.nl (K.O.); fjanssen@iclon.leidenuniv.nl (F.J.)

[2] Research Institute of Child Development and Education, University of Amsterdam, 1012 WX Amsterdam, The Netherlands; C.A.M.vanBoxtel@uva.nl

\* Correspondence: m.dam@iclon.leidenuniv.nl

**Abstract:** Out of all the complex systems in science education curricula, cellular respiration is considered to be one of the most complex and abstract processes. Students are known to have low interest and difficulties in conceptual understanding of cellular respiration which provides a challenge for teaching and learning. In this study, we took literature about modelling and teaching and learning cellular respiration as a starting point for the design of a concrete dynamic model in which students (n = 126) use Lego® to simulate the process of cellular respiration. Students used the simulation embedded in the context of finding the efficiency of a sediment battery as a future source of green energy and we tested the effects on conceptual learning and situational interest in an experimental study. Results on conceptual learning show that both experimental and control groups had comparable results in the test. The questions that students in the experimental group asked during enactment, however, gave notice of a focus on both isolated component parts as well as modes of organization at higher organizational levels which is linked to how biologists mechanistically understand complex systems. Both groups report a similar high measure to which the topic is meaningful in real life (situational interest value), whereas the enjoyment (situational interest feeling) was significantly increased in the experimental group. Furthermore, students report specific advantages (e.g., I now understand that one acid chemically changes into another and they do not just transfer atoms) and disadvantages (e.g., time issues).

**Keywords:** cellular respiration; concrete dynamic model; Lego®; simulation

## 1. Introduction

The explicit study of complex systems has increasingly received attention in science over recent decades. Complex systems are defined as systems with multiple interacting levels and a hierarchical nature where the function is not so much the result of isolated components, but rather of decentralized interactions of multiple components over the organizational levels, sometimes referred to as emergence [1,2]. As phenomena in the sciences are often complex and not directly visible, models covering both macro and sub-micro levels of representation play an important role in the core objectives of science, being comparisons, predictions and/or explanations about phenomena [3]. Models play an important role in the production, dissemination and acceptance of knowledge [4].

As many of the phenomena studied in science contain highly dynamic elements and are not directly visible to the naked eye, scientists are known to make abundant use of simulations to make predictions, comparisons and aid understanding. Simulations have a long history, originating from

fields such as meteorology and nuclear physics, after which they have become indispensable in recent decades in a growing number of disciplines such as engineering, climate science, evolutionary biology, ecology, economics, medicine, and many others. A simulation can be described as an imitation of a process or phenomenon for which a dynamic model is used. The dynamic model represents the key characteristics of the process at hand and the simulation is performed to understand or predict the operation of the process over time. The term 'simulation' is often used interchangeably with 'computer simulation' [5–7]. However, as simulations can have many forms and variations [8], interchangeable use of these terms is not advocated. Rather, in the same way as scientists develop and apply simulations to model and understand specific natural phenomena, we define simulations as those imitations and models that specifically help imitate and convey the process at hand best, whether done by using the more simple paper-pen or cut and paste models, more elaborate concrete models such as 3D-models, using computers, or otherwise. As simulations are abundantly used in science, it follows naturally that they should play equally important roles in science education [3,9]. In science education, the importance of simulations has been stressed for teaching and learning complex and highly dynamic processes [8]. Simulations permit fast manipulation of many components and variables and learners receive immediate feedback from the simulation. In a review about using computer simulations in science education, Rutten et al. [7] found that computer simulations can enhance motivation and attitude in most cases and lead to learning gains in specific cases, for example when laboratory activities are replaced by simulations. However, as simulations come in great variety in both form and content, comparison between different settings is rather hard [5].

Simulations are nowadays widely used in science education, with a special emphasis on their use in physics. In physics education, simulations are mostly used to study the outputs of complex systems resulting from certain input variables and by interactively changing conditions, variables or courses of action (laws), output can be adjusted [10–12]. In biological education, however, complex systems are made up of a numerous amount of parts on many different, yet interwoven organizational levels ranging from microscopic to macroscopic levels and often without central control [11–14]. Besides cellular respiration, other examples of such complex biological systems are the circadian rhythms that occur in many physiological processes and behavior, the systems that determine how genes determine traits in molecular biology, and processes such as immunology and photosynthesis.

An important question, then, becomes how we can design and deploy simulations specifically aimed at assisting students to gain understanding of complex biological systems. In this study, we design a simulation using a concrete dynamic model for which we first distill design principles from literature and test the extent to which students gain conceptual understanding of one of the most complex and difficult biological topics, being cellular respiration. Because of the novelty of this approach, we chose to combine quantitative measures with qualitative measures in which we follow how students' conceptual learning takes place and how they evaluate the simulation.

## 2. Theoretical Framework

### 2.1. Teaching and Learning Cellular Respiration

Cellular respiration is a biological process that consists of a sequence of subprocesses in the eukaryotic cell plasm and mitochondria. The process starts with nutrients such as glucose and results in the production of adenosine triphosphate (ATP), which are molecular units for energy transfer in organisms. For a fully elaborated description of the process, see Appendix A.

Earlier research has shown that there are some common difficulties in teaching and learning cellular respiration. Both teachers and students stated it to be one of the most complex and hardest topics in biology and student interest in the topic is known to be one of the lowest out of the biology curriculum [15–18]. The question as to why it is considered to be so complex and hard can be partly answered by the rationale that the entire process is considered to be highly abstract as a result of not being visible to the unaided eye (micro level) with little connection to functions on higher organizational

(macro) levels and that it contains many distinctive steps that function in a complex system which make it hard to follow. Second, it is also found to be difficult to relate the subprocesses to each other and to real life [15]. Third, the multitude of details and the need of a new vocabulary make it a challenging topic [18,19]. Fourth, students have also been shown to have some common misconceptions. A first misconception is that arrows in static flow diagrams always display real physical movement of parts instead of parts changing shape or binding to receptors [20]. This misconception is the result of a low visual literacy and reinforced by the abundant use of static flow diagrams in biological textbooks. A second misconception is that cells should need oxygen for respiration in the same way as humans do on the organismic level [21]. This misconception stems from the fact that students have to study cellular respiration in simplified form in lower grades. This overarching simplification leads to the misconception that implies that all subprocesses require oxygen and that without oxygen, cells do not produce energy.

From these characteristics, we can derive that a dynamic model for cellular respiration should make the process visible for the unaided eye, it should allow students to actively explore the process by working with a multitude of details, stimulate interest, allow students to work through the entire process instead of isolated facts and terms and, finally, it should connect processes on microscopic and macroscopic levels and make connections to real life.

### 2.2. Modelling Cellular Respiration

The demands as informed by the topic of cellular respiration can be roughly described according to four aspects. First, we know that cellular respiration is one of those phenomena in biology that is either too small, too large, too fast, or too slow to see or experience with the unaided eye. Especially for these phenomena, representations that focus on visualization are considered to be important, which relates to two of Gilberts' [22]. overview of modes: the concrete mode and the visual mode [23,24]. Second, as students often have low levels of prior knowledge about the process of cellular respiration and the process itself has a large amount of interactions and parts that move and change shape in several subprocesses, a static visual model is considered inadequate. Rather, a dynamic model consisting of many concrete manipulable elements does fit the process of cellular respiration better. A dynamic model allows learners to interact with the materials and explore the process based on their assumptions and prior knowledge. Also, a dynamic model is hypothesized to play an important role by making links between macroscopic and molecular scales [19,25]. Third, as student have low interest in the topic, a model that is both concrete and dynamic could influence students' affective outcomes. Already in 1968, Maslow described manipulating a concrete model as a specific form of internal delight, which refers to the actualization of self. This expected internal delight might influence student interest positively. In Maslow's [26] words:

> From the moment the package is in his hands, he feels free to do what he wants with it. He opens it, speculates on what it is, recognizes what it is, expresses happiness or disappointment, notices the touch of the product, the different weight of the parts, and their number and so on ... There is physical, emotional, and intellectual self-involvement; there is recognition and further exploration of one's abilities; there is initiation of activity or creativeness.

—[26] (pp. 49–50)

Concrete or physical models can convey spatial relationships and mechanisms in unique ways. Work in the field of molecular cell biology shows that concrete models can activate perceptual and cognitive processes that go beyond the visual and bring a sense of reality and natural interaction into the process of exploration and understanding [27]. Fourth, as the topic was found to be difficult to relate to real life, the concrete dynamic model by which simulation takes place should also be embedded in an authentic context, see Section 2.5.

## 2.3. Scaffolding and Structuring the Modelling Process

From literature it is known that simulations using a dynamic model require rather open learning environments. However, they also require certain scaffolds and structures to be used effectively [5]. Structuring the modelling process relates to issues about sequencing and complexity. One of the important considerations is if students are allowed to see the entire simulation at once, or rather leave out certain parts or variables because of the overwhelming complexity and potential frustration. Lazonder and Kamp [28] tested the effects of offering a complete simulation versus a gradual build up, resulting in higher conceptual learning for the second group. This resembles the setting of playing a computer game moving through the game in increasingly complex levels to avoid frustration. The first design principle, therefore, is to gradually open up the content of cellular respiration in the simulation and assist students to gradually explore the subprocesses of the simulation before exploring the entire process. A second design principle for complex systems where prior knowledge is low is to allow students to actively explore the difficult conceptual area at hand by building on their prior knowledge. Low levels of prior knowledge require more guidance in the materials and vice versa [29,30]. This means that we should not present students with a fully expressed concrete dynamic model for cellular respiration, but rather lead students towards understanding the entire process via intermediate self-generated modelling activities [31,32].

Providing scaffolding is important because highly interactive models can impose an undesirable high-intrinsic load [33]. Attention should be given to a correct level of instructional guidance as it is known that not only too much, but also too little guidance can hinder learning [23]. De Jong et al. [5] emphasize the specific importance of guidance in using dynamic models and state that this should have the form of prompts, heuristics and scaffolds. Prompts are considered to be questions (e.g., 'provide an explanation at this point' or 'What is . . . '), other concise stimulations to bring about acts (e.g., 'action is required at this point') and examples (e.g., embedding worked out examples or fill in the blanks exercises). Heuristics can be defined as fixed productive ways of acting within the simulation in gradually increasing difficulty. Scaffolds come in many shapes and sizes, as long as they provide structure and assistance for the learners on a larger grain size as they move through the simulation. In simulations for complex systems such as cellular respiration, we propose that already developed schemas for guidance can serve as a scaffold to assist the learner throughout the simulation [34]. Finding such a scaffolding schema is further informed by the highly dynamic topic of cellular respiration.

But what is it that drives modelling? We know that having an appropriate structure and a correct amount of instructional guidance in the form of structure, prompts, heuristics and scaffolds can guide and assist the modelling process [5]. But the very act that drives the quest for knowledge expansion and initiates and sustains the modelling process is the student questioning which is so very fundamental to science and scientific inquiry [35,36]. Structures, prompts, heuristics and scaffolds during the modelling process should lead students to asking questions that will guide them through the modelling process and at the same time find information to understand the complex systems.

A final aspect in the modelling process is the use of additional models or representations while using simulations. Literature shows that using multiple models or representations in simulations can have specific advantages over using a single one [5,37]. Representation forms such as diagrams, tables, texts, multimedia animations or video each have their own benefits, depending on the process and setting involved. Ainsworth [38] presented a sound framework to distinguish three roles that offering multiple representations might have: a complementary role, the role of constraining misinterpretations, and the role of creating a deeper understanding due to relations and extensions. In most cases, designers aim for complementary roles for models as this aids deeper understanding [32].

## 2.4. Contextualizing the Modelling Process

As cellular respiration was found to be hard to relate to real life, we embedded the simulation in a context. Context-based education is an approach to education in which subject matter is organized and taught by using contexts [39]. At classroom level, a context-based lesson starts with presentation of a

context, from which a problem or question logically follows. Next, interactive student learning activities within the context setting lead students to answering the central question or solving the problem. While students perform the learning activities, they learn to understand one or more biological concepts that focus on the learning goals set beforehand. In the final phase, reflection on the answer or solution takes place and in some cases, relevant biological concepts are decontextualized [40,41]. In this study, students used the simulation in the second lesson phase, which was preceded by a context introduction with attending assignments and followed by a lesson phase in which they answered the assignment flowing from the context (see Figure 1, right column). The more a context becomes authentic, the more complex the investigations and explanations will be [40]. Therefore, the choice of a context being authentic or not depends on the complexity of the topic at hand, the levels of prior knowledge, the expected student interest and the time allotted for science education in the school curriculum.

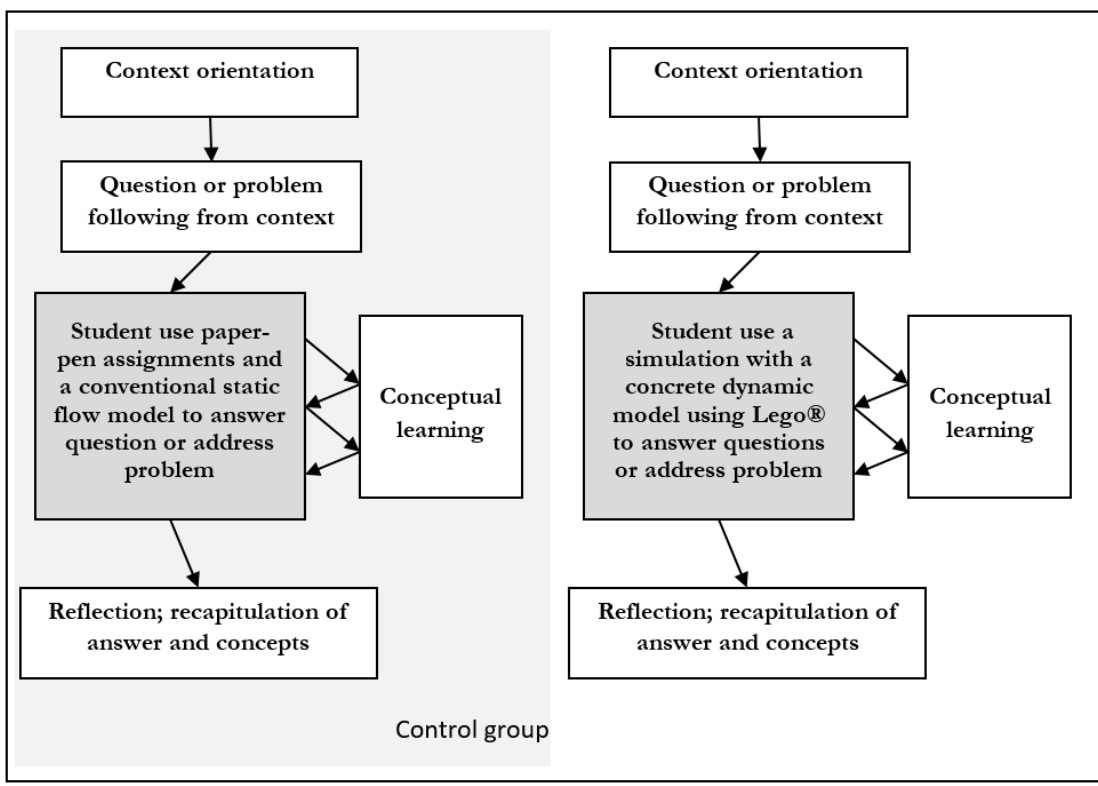

**Figure 1.** Survey of experimental research design. Note. © Adapted figure based on Wieringa, Janssen and van Driel [41]. Used with permission.

*2.5. Student Interest*

Student interest in cellular respiration, as described earlier, is one of the lowest in the biology curriculum. In more general terms, student interest in science was found to be low and students often have negative attitudes towards science [42]. Student interest and fostering a positive attitude are also rarely targeted in science classes as students in science mostly listen to lectures and carry out standard "cookbook" laboratory activities [8]. In recent years, the focus of educational research on affective constructs such as interest has increased. Hidi and Renninger [43] have proposed a widely accepted model that presents student interest in two major forms: individual interest and situational interest (SI). Situational interest is interest that is generated by aspects of a specific situation and individual interest is a longer-term preference for a particular subject area or domain. In the present study, we focused on the introduction of a simulation for cellular respiration in a specific situation, which relates to situational interest. This construct of situational interest is made up of three variables, being triggered-SI, maintained-SI-feeling and maintained-SI-value [44]. In comparison to triggered SI which speaks of the initiated interest in peripheral aspects of the materials, the two maintained SI forms speak

of a more involved and deeper form of SI because the enjoyment is based in the domain content [44]. Earlier work on SI has shown that maintained-SI can easily develop into individual interest, but there are also some differences between maintained-SI and individual interest. For instance, maintained-SI develops in response to exposure to material in a particular setting, whereas individual interest results from repeated exposure to materials across settings [44].

## 3. Research Questions

The leading research question for this study was stated as follows: what are the effects of simulating cellular respiration within context using Lego® as a concrete dynamic model? We split this research question into four sub-questions:

1. What are effects on conceptual learning?
2. What are effects on situational interest?
3. How does conceptual learning take place during enactment?
4. How do students evaluate the simulation of this complex biological system?

To find answers to the research questions, we used a combination of quantitative and qualitative research methods. For quantitative outcomes on conceptual learning and situational interest (research questions 1 and 2), we designed an experimental study in which we compared an experimental group that performed a simulation using a concrete dynamic model to a control group that performed conventional paper-pen assignments using a conventional static flow diagram. Our hypothesis for this is that the experimental group will show that they gained better understanding of cellular respiration compared to controls by showing increased signs of conceptual learning. Also, we expect students in the experimental group to report higher levels of situational interest. For qualitative outcomes (research questions 3 and 4), we used a think aloud protocol, an open questionnaire, and took field notes with the goal of finding out how student learning takes place and how they evaluate the simulation.

## 4. Methods

### 4.1. Concrete Dynamic Model Design

We chose to design a concrete dynamic model consisting of Lego® bricks, wheels and plates and had students build carts that drive around Lego® road plates. From early childhood, many children are expected to have experience in playing with Lego® carts which can serve as an already developed schema. Also, as the dynamics of working with the bricks, wheels, plates, colors, sizes, et cetera is expected to be an appropriate scaffold considering the ubiquity of microscopic parts and dynamics of cellular respiration. Students visualized the model parts at the grain size of molecules and atoms. Relating to the colors often used in chemistry education, carbon was defined to be a black Lego® brick (size $1 \times 2$), oxygen a red brick and hydrogen a white brick. Next, students were asked to assemble a glucose molecule ($C_6H_{12}O_6$) and put it on a thin Lego® plate (size $4 \times 10$) attached to wheels to make it a small cart that can move around. These self-generated molecule-carts were placed on a road map (see Table A2, Appendix C) in order to simulate the process of molecular movement in the different subprocesses of cellular respiration.

In the model, we presented students with several scaffolds and prompts (see Table 1). The first scaffold is that students receive a car route depicting one of the subprocesses of cellular respiration using printed imaged of Lego® road plates. On the printed images, we included prompts in the form of boxes alongside the road where action is needed. In every box, a molecular change is prompted (e.g., ATP meaning that something happens to ATP, $CO_2$ meaning that the molecule releases or binds to $CO_2$). A second scaffold is that students receive a box filled with Lego bricks, wheels and plates that can be used to construct a cart. A third scaffold is presenting students with a legend of what the colored Lego bricks mean (e.g., red = oxygen, white = hydrogen). A fourth and final scaffold of the concrete dynamic model is that students receive the conventional static flow diagram prescribed by the

Dutch curriculum on paper (see Table A1, Appendix B). Students are allowed to use this conventional model and attending handbook [45] during final exams.

**Table 1.** Survey of the context and simulation for experimental and control group.

| *Experimental Group* | | |
| --- | --- | --- |
| Lesson Phase 1 | Lesson Phase 2 | Lesson Phase 3 |
| Introduction of the context and attending assignment | Simulation of the process | Finding the efficiency (answering the assignment flowing from the context) |
| *Content*<br>Teacher presents students with an authentic story of how researchers study the use of prokaryotes for energy production in sediment batteries. Students get the role of researchers wanting to find the efficiency of such sediment batteries for green energy and use Lego as a simulation to do so. Main assignment is the question: how much sugar is needed to create a current that resembles the amount of energy yielded by a single AA battery (1.5 Ah battery)? | *Scaffolds*<br>1. Students receive a printed car route depicting a sub-process of cellular respiration [a] using printed imaged of Lego® road plates<br>2. Prompt: Every step in the molecular change pathway is depicted as a car stop with molecule reference to imply that action is needed<br>2. Students receive a box of Lego bricks including wheels and plates<br>3. Students receive a legend of what the colored Lego bricks represent (e.g., red = oxygen, white = hydrogen)<br>4. Students receive the conventional static flow diagram on paper | *Scaffolds*<br>1. Students receive information about the amount of electrons yielded upon oxidation by the two electron acceptors in the process (NADH and $FADH_2$)<br>2. Students receive information about the necessary conversion formula (1 Ah is 3600 Coulomb; the molar mass of glucose $(C_6H_{12}O_6) = 180{,}1559$ g/mol; mass (m) = M × n; Avogadro's number) |
| *Control Group* | | |
| Lesson Phase 1 | Lesson Phase 2 | Lesson Phase 3 |
| Introduction of the context and attending assignment | Assignments to find information | Finding the efficiency (answering the assignment flowing from the context) |
| *Content*<br>Same as test group | *Scaffolds*<br>1. Students receive the conventional static flow diagram on paper<br>2. Students receive multiple questions that guide them towards the answer (e.g., find out how many ATP, NADH and $FADH_2$ are used and formed in each sub-process/what are the names of the sub-processes in correct order?) | *Scaffolds*<br>Same as test group |

[a] In order to get a full picture of the entire process, students use their simulation of the sub-process halfway through the lesson to explain this to another group that has a complementary sub-process and vice versa (either glycolysis plus decarboxylation or the citric acid cycle).

### 4.2. Context Design

For this study, we designed a context based on one of the elements described in the Dutch final exam requirements which makes reference to the metabolism of prokaryotes and their applicability in a biotechnological context. For this, we chose the context of a research team wanting to find the efficiency of using a sediment battery as a source of alternative energy in this time of anticipated shortage on fossil fuels. In a sediment battery [15,46], bacteria (prokaryotes) are used for production of energy.

In anaerobic sediment, bacteria can perform cellular respiration in which electrons are being transferred to electron carriers and the electron transport chain. However, as bacteria do not have mitochondria, the electron transport chain will be performed along the membrane of the bacterium. Using graphite electrodes, one can harvest the electrons from this membrane. The graphite electrode can then be connected to both the anaerobic sediment (anode) and to aerobic water where electrons are taken up by oxygen (cathode). This establishes a flow of electrons, producing a current [47]. The objective for students in the context was to find the efficiency of such a current by calculating how much sugar (in grams) is needed to produce a capacity of a single AA battery (1.5 Ah).

### 4.3. Lesson Phases and Enactment

A survey of the lesson phases is presented in Table 1. In lesson phase 1, we presented the context and attending assignment after which students simulated the process of cellular respiration in lesson phase 2. Students paired in groups and first received one of the subprocesses of cellular respiration (glycolysis or decarboxylation combined with the citric acid cycle). The objective was to first proceed through all the steps of the allotted subprocess by removing, adding or changing the shape of the Lego® 'molecule' on the cart. If the final product and all intermediate products were accurate, students explained their subprocess by means of demonstrating the simulation to another group who had the complementary subprocess. Next, they could proceed to lesson phase 3 in which they solve the assignments following from the context of counting all electrons taken up by either FAD or NAD and use this number to calculate the efficiency of the battery by calculating how much sugar (in grams) is needed to produce a capacity of a single AA battery (1.5 Ah) or explain the process to others (depending on the context). For lesson phase 3, we provided scaffolding in the form of information about both the amount of electrons yielded upon oxidation by the two electron acceptors as well as conversion formula needed for calculation (see Table 1). In the control group, students also worked in pairs and lesson phases one and three were the same as the experimental group. In the second lesson phase, during the time in which the experimental group performed the simulation, the control group received a list of questions on paper that guide them towards finding as answer in a way that many teachers would normally do (see Table 1).

### 4.4. Research Design and Participants

In this study, we combined quantitative and qualitative research methods. We used an experimental research design for answering the research questions about the quantitative effects of the intervention on situational interest and conceptual learning. For this experimental study, students (n = 126) were randomly assigned to either experimental (n = 76) or control group (n = 50) within classes to ensure adequate comparable groups. The control group performed conventional paper-pen assignments and used a conventional static flow diagram. See Figure 1 for the differences between experimental group and control group treatment.

For answering the research questions about qualitative effects (student questioning and student evaluation) we used three qualitative research methods: think aloud protocols, an open questionnaire for measuring advantages, and disadvantages and taking field notes. In the think aloud protocols (n = 8) two groups from the experimental group expressed their thoughts out loud for each sub process during classroom enactment. We selected students for these two groups randomly from two different schools. We administered the evaluative open questionnaire for advantages and disadvantages of the simulation to those students from the experimental group that were available due to not having a lesson directly after the ones by which this research was performed (n = 36).

The study was performed in upper pre-university education classes in the Netherlands (16–18 years old). Students were distributed over seven classes stemming from five schools in the northwest of the Netherlands. The highest class size was 31 and the lowest 18. All schools were public schools and were located in a radius of approximately 40 km from the researchers' institute in the West of the Netherlands within densely populated areas (>1000 per km$^2$). Contact with these schools was

established via the contact details of biology teachers that attended a seminar about modelling in biology education a few months earlier and gave notice of interest in participating in future research. Following that, the teachers of the students involved in this study were interested in model use in biology education.

### 4.5. Instruments

Conceptual learning test. Conceptual learning was measured by using a 7-item test that was constructed on the basis of Dutch final exam requirements. For this test, we took the question formulation as designed by Patro [18] as a starting point and elaborated these with the Dutch final exam requirements in mind. Examples of item formulation in the test were to state the correct sequence of the subprocesses of cellular respiration, or to mention how many ATP molecules are used and generated for glycolysis. We first piloted this test by having two experienced biology teachers and three students of the same age and school type as the target group make comments, after which they made adjustments on issues like phrasing and clarity.

Situational interest questionnaire. To study the effects on situational interest, we used questionnaires based on work by Linnenbrink-Garcia et al. [44] which we translated. We measured two variables for maintained situational interest (see also Section 2.5). The first variable relates to feeling-related components which we called SI-F and the second variable relates to value components which we called SI-V. The SI-F component characterizes individuals' affective experiences while engaging with domain content (e.g., enjoyment, excitement). Example of an item for measuring SI-F: 'I am excited about these lessons'. The SI-V component, on the other hand, emerges as individuals come to believe that a certain topic is important and meaningful to them. Examples of items for measuring SI-V are: 'What we are studying in these lessons on cellular respiration is useful for me to know' and 'The topic of these lessons can be applied to real life'. Students scored the items on a 5-point Likert Scale (totally disagree—totally agree). Cronbach's alpha for the total scale score of SI-F (three items) was found to be 0.84 and for the total scale score of SI-V (four items) 0.74.

Think aloud protocol. In the think aloud protocol, students from the experimental group were asked to express everything they thought out loud during lesson phases 2 and 3. Recordings of these thoughts were made using audio recording equipment.

Student evaluation. We asked students (n = 36) to individually express any advantages and disadvantages they could think of directly after working with the simulation. For this, we used an open question format.

Field notes. Finally, as the first and second author visited all classroom enactments, we also performed classroom observations and took the resulting field notes into account when describing results as field notes are widely recommended in qualitative research as a means of documenting needed contextual information [48].

### 4.6. Data Analysis

Conceptual learning test. The mean outcome (on a 1–7 Likert scale) of the experimental group (n = 76) was compared to the control group (n = 50). We tested this using a two-side independent-sample *t* test.

Situational interest questionnaire. The total scare score of the experimental group (n = 76) was compared to the total scare score of the control group (n = 50). A total scale score is the sum of the individual item scores for each construct (SI-V or SI-F). We tested this using a two-side independent-sample *t* test. Cohen's d was determined as an effect size.

Think aloud protocol. First, the audio recordings were transcribed verbatim. The first and second authors then independently scored the transcripts and isolated all questions the groups asked while performing the lesson phases (simulation and finding the efficiency). We then analyzed these questions, which mostly meant one sentence per analysis unit. We based our analysis on the ways in which biologists seek explanations for complex systems. Biologists often work mechanistically in discovering

how a mechanism works by a reductionist inquiry which requires decomposing the mechanism into its parts and next, seeking to recompose the mechanism to fulfill its complex function. Because of such an approach, we chose to analyze the student questions based on the difference between parts or entities on the one hand (lower level) and modes of organization or purpose on the other hand (higher level). Such a framework has earlier been proposed by Russ, Scherr, Hammer and Mileska [49] who distinguish between describing and identifying entities as the lower level of mechanistic reasoning, and organizational structures (interactions) and purpose of organized entities as the higher level. We do not use explicitly the full framework that was presented by Russ et al. [49] because our focus lies more on student questioning and not so much on student talk or reasoning in a broader sense. Thus, when questions expressed words pointing towards single entities or how to build single entities, forms or single parts, they were classified in the left column: questions focused on entities (lower level). Examples of such questions are: 'What is the chemical structure of the starting molecule'? or 'how shall we build the two C3 carts'? When questions made reference to the organization of entities or purposes of organized entities, they were in the right column, being questions focusing on interactions or purpose (higher level). Examples are: 'What is the purpose of acetyl-coA binding to pyruvate'? or 'where do electrons stored in sugar go in the process'? In case of disagreement, first and second author discussed until agreement was reached.

Student evaluation. The first and second authors analyzed the student statements by grouping those statements that referred to the corresponding themes. The relative amount of the statements was perceived as high, intermediate or low by making distinction between those occurring at fractional factors under 0.3 (low), 0.3–0.6 (intermediate) and above 0.6 (high).

Field notes. We analyzed the paper-pen field notes by classifying them into the following categories as proposed by Phillipi and Lauderdale [50]: a. Field notes about study context (seasons of data collection, geographic setting, classroom sizes), b. Participants' non-verbal behavior c. Teachers-student interactions and how teachers made affective comments during and following enactment.

## 5. Results

### 5.1. Conceptual Learning

We found no significant effect on conceptual student learning when comparing experimental group ($M = 4.79$ on a 1–7 Likert scale; $SD = 1.0$) and control group ($M = 5.0$; $SD = 0.99$; $t = -1.362$; $df = 124$; $p = 0.173$), see Figure 2.

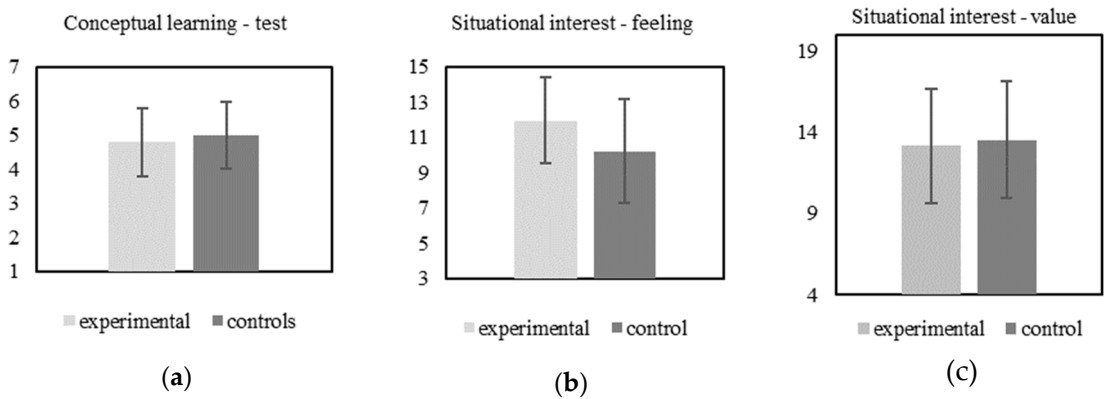

**Figure 2.** Survey of experimental outcomes. (**a**) For conceptual learning, the means and standard deviation (SD) are presented; (**b**) For SI-feeling (three items) the total scale scores and SD's are presented; (**c**) For SI-value (four items) the total scale scores and SD's are presented.

## 5.2. Situational Interest

The scale scores for situational interest-feeling (SI-F) indicate that this was perceived as relatively high in the experimental group (scale score = 11.97; $SD$ = 2.43). This SI-F mean was found to be significantly higher than for controls (scale score = 10.22; $SD$ = 2.94; $t$ = 3.64; $df$ = 124; $p$ = 0.001). Cohen's d equaled $d$ = 0.66 suggesting a medium effect size, see Figure 2

The scale scores for situational interest-value (SI-V) show rather high scores for both experimental group (scale score = 13.17; $SD$ = 3.53) and control group (scale score = 13.54; $SD$ = 3.59; $t$ = −0.57; $df$ = 124; $p$ = 0.572). A one-sample t-test was carried out for the scale scores of SI-V ($M$ = 13.32; $SD$ = 3.54 $t$ = 4.168; $df$ = 125; $p$ = 0.000) indicating that the scores differ statistically significant from the null hypothesis, middle of the scale. No significant main effects were found when comparing the total scale scores for SI-V in the experimental and control groups, see Figure 2.

## 5.3. Student Questioning While Thinking Aloud

We present the results from the two groups that performed the simulation while thinking aloud in Table 2. From this survey, it becomes clear that students ask many questions, some of which are focused on entities (e.g., What happens in the first step to the glucose molecule?) referring to lower levels of mechanistic reasoning. Some of the questions, however, focused on interactions or purpose (e.g., 'How do these acids actually change or transfer atoms') referring to higher levels of mechanistic reasoning.

**Table 2.** Questions asked during enactment.

| Lesson Phases | Questions Focused on Entities | Questions Focused on Interactions or Purpose |
|---|---|---|
| Phase 1—Context introduction | None | None |
| Phase 2 [a] Simulation of the sub-process glycolysis and decarboxylation | - How will we assemble the car?<br>- What is the name and chemical structure of the starting molecule?<br>- How do we build glucose with Lego®bricks on the car?<br>- What happens in the first step to the glucose molecule?<br>- What happens to the phosphate released from ATP, does it bind to glucose or is it lost?<br>- Does NAD receive one or two H+-protons?<br>- How many C-atoms are left at the end of decarboxylation?<br>- So how many NADH are formed in glycolysis when the second part takes place twice? | - Why do these first steps of glycolysis cost ATP, the process is aimed at generating ATP and energy, right?<br>- The glucose (C6) molecule splits in two C3 molecules, does this mean that the second part of the process takes place twice?<br>- The second part of glycolysis takes place twice, does this happen sequentially or parallel to each other?<br>- So the purpose of glycolysis and decarboxylation is to create pyruvate, where is the energy from glucose stored? |
| Phase 2 [a] Simulation of the sub-process citric acid cycle | - What is the name and chemical structure of the acid that 'receives' the remaining of pyruvate bound to Acetyl-coA?<br>- The second acid is citrate, is this why it is called the citric acid cycle?<br>- How can GTP formation lead to ATP production?<br>- Does NAD receive one or two H+-protons?<br>- Does FAD receive one or two H+-protons?<br>- - Do we have oxaloacetate as the residual molecule that can re-enter the cycle? | - How do these acids actually change or transfer atoms?<br>- What makes these acids change all the time?<br>- Why is water first added and then directly split off again?<br>- If all the carbon atoms are just split off to form $CO_2$ all the time, why does glucose generate so much energy?<br>- This sub-process only yields two ATP, where is the rest produced?<br>- Where did all the energy from glucose go? |

**Table 2.** *Cont.*

| Lesson Phases | Questions Focused on Entities | Questions Focused on Interactions or Purpose |
| --- | --- | --- |
| Phase 3<br>Finding the efficiency | - Does a glucose molecule actually release 24 electrons, or does it contain more?<br>- How do I calculate with the formula for the amounts mol and molar mass? | - Do all electrons get captured by the graphite? Is it 100% efficient?<br>- Are all electrons similar in energy amount?<br>- Why don't we use this method for our energy production if we only need 0.4 grams of sugar for an AA battery? |

[a] In order to get a full picture of the entire process, students use their simulation of the sub-process half way through the lesson to explain this to another group that has a complementary subprocess and vice versa (either glycolysis plus decarboxylation or the citric acid cycle).

*5.4. Student Evaluation*

Students identified specific advantages of using the concrete dynamic model over a static flow diagram (see Table 3). A majority of students stated process advantages such as: "I can better see the process of specific molecules changing in the process" and "It presents me with a better overview over the entire process from a glucose molecule to ATP". Also, discovery- and interactivity advantages were also abundantly present. The most frequent disadvantages of the concrete dynamic model over a static model stem from the category time, e.g., 'the Lego® model takes longer to apply'.

**Table 3.** Survey of advantages and disadvantages.

| Category | Sub Categories | Rate * |
| --- | --- | --- |
| **Advantages** | | |
| Processes | I can better see the process of specific molecules changing in the process<br>I now understand that one acid chemically changes into another and they don't just transfer atoms<br>I now understand that a phosphate groups released from ATP molecules are being reused somewhere else in the process<br>I now better understand why a part of glycolysis and the Krebs cycle are being executed twice | High |
| Discovery and Interactivity | I immediately took notice whenever I made mistakes<br>I liked it that I could discover the process myself instead of explanation by the teacher | High |
| Multiple representations | Using both the Lego® model and the static textbook model, I now better understood the static model | Intermediate |
| Mental model | I can easily remember this as a model in my memory and in a test I will think back to these little cars | Intermediate |
| Visual aid | The colors and images of the plate and bricks helped me to learn and gain deeper understanding | Low |
| **Disadvantages** | | |
| Time | The Lego® model takes longer to apply | Intermediate |
| Use at test | The static model can be studied or even used during a test, the Lego® model can only be done | Low |
| Difficulty | It can be quite hard to visualize and build all molecules | Low |

* Note: The relative amount of the statements was perceived as high, intermediate or low by making distinction between those occurring at fractional factors under 0.3 (low), 0.3–0.6 (intermediate) and above 0.6 (high).

*5.5. Field Notes*

In regard to the study context, we visited the five schools during the time that cellular respiration was scheduled in the curriculum, which is very diverse over school curricula in the Netherlands. We visited three schools in the winter, one in the autumn and one in the spring. During enactment, we noticed that students payed close attention during context introduction and it became clear that none of the students were familiar with the sediment battery as a source for green energy. When students

received their Lego® boxes in pairs, we could observe their excitement as many started smiling and chatting about earlier experiences with Lego® bricks. Lots of students stood up working with the concrete dynamic model and did not sit down for the first 15–20 min. In the cooperation setting, students could choose who to pair up with and we did not observe any disrupting behavior during the enactment. In two classes, students were given a short break (5 min) at half time. It became clear that the time on task during the simulation was very high. We did not observe major off task behavior, besides sometimes the inescapable school chit chat.

During enactment, teachers expressed their role change away from the instructor towards becoming a coach, a role which teachers liked for this topic. Also, they said that it is hard not to present an answer to student questions directly, but to stimulate students to find the answers themselves. All six teachers wanted to buy the set for themselves when students were done, something we accounted for by presenting them with a list of necessary bricks and other materials and places to order them from. As all biology teachers gave notice of interest for participation during a seminar about biological modelling, they were already interested in biological modeling prior to the study. Working with this simulation based on inquiry processes in their own classes was a discovery process for them as well, as they never had tried something like this before. Teachers expressed that they saw joy entering their classroom and unexpected student humor (e.g., when students accidently dropped the paper board, students said: 'oops, there goes your decaboxylation'). In a regular, lecture-based teaching situation, some teachers felt a burden to help students understand the static flow diagrams. However, students using this simulation discovered how to understand the conventional static flow diagram in 3D (see Table 3, 'Using both the Lego® model and the static textbook model, I now better understood the static model') which takes away many of the misconceptions.

## 6. Conclusion and Discussion

In this study, we focused on teaching and learning complex systems by means of simulation and tested the effects on conceptual learning and situational interest. To this end, we chose one of the most difficult and complex biological systems, cellular respiration, and designed a concrete dynamic model in which students use Lego® bricks, wheels and plates to create a cart that simulates the process of cellular respiration driving along paper road plates assisted by scaffolds, heuristics and prompts for action. We embedded the simulation in the authentic context of finding the efficiency of a sediment battery for green energy production.

Results show that the simulation for cellular respiration seems to affect conceptual learning in two ways. The simulation leads to similar conceptual learning compared to a conventional static flow diagram when measured in a post-test. However, the questions that students in the experimental group asked during enactment of the simulation were shown to be both lower level questions focusing on microscopic entities and the structure of the concrete task as well as higher level questions focusing on interactions of the entities, modes of organizations on higher levels and the purposes of combinations of subparts and subprocesses (e.g., 'So the purpose of glycolysis and decarboxylation is to create pyruvate, where is the energy from glucose stored?'). Together with the advantages expressed once the simulation was done (e.g., 'I now understand that one acid chemically changes into another and they don't just transfer atoms' or 'I now better understand why a part of glycolysis and the Krebs cycle are being executed twice') these results hint at triggering thought processes that follow the ways in which biologists see complex systems as being the result of decentralized interactions between subparts on interwoven organizational levels ranging from microscopic to macroscopic levels.

Results from measuring situational interest show first that the students in the experimental group found their materials more enjoyable and exciting than the control group (situational interest-feeling). Also, both groups valued the meaning and importance of the topic highly for real life. We found no significant change for this value between the groups (situational interest-value). This indicates that the use of a context itself can stimulate feelings of the topic being meaningful to real life and not so

much the design of student activities by which students find answers to the questions following from the context.

From the outcomes of student evaluation measures, we can conclude that students from the experimental group have a rather positive attitude towards learning the ins and outs of the process over time as well as the ways of learning while simulating (discovery and interactivity, e.g., 'I liked it that I could discover the process myself instead of explanation by the teacher'). They expressed these advantages even while also seeing specific disadvantages (intermediate occurrence) on time issues ('The Lego® model takes longer to apply') when compared to the static model. However, given the fact that teachers, in most cases, have limited teaching hours and multiple goals to address at the same time, these time issues and feasibility of using several simulations in the curriculum is important to address. In this study we aimed to design the simulation as practical for teachers as possible by, e.g., using tailored amount of Lego® bricks, offering printable road plates and making explicit how exam requirements are covered so that the simulation can replace regular lessons, but more research is needed on how simulations for several complex systems can be made practical for teachers.

From the field notes, we can conclude that there was a sense of joy or excitement present in the classrooms as well as a value for real life, as we observed behaviors that hint towards significant time spent on task and an eagerness to model and simulate the process in order to understand the ins and outs of the process and find an answer to the questions presented in the context.

In regard to literature about modelling in simulations and the limitations in our research, a first thing to notice is that the simulation in this study used a concrete dynamic model, which is different from most simulations in which dynamic computer models are used [5]. Also, as we chose to focus on a complex biological system, the dynamic model was different from most other dynamic models that focus more on complex systems in physics or chemistry and study output as a result of adjusting variables, changing conditions, and more or less fixed heuristics or courses of action or laws within the system [11–13]. Second, students in our study used a concrete dynamic model to recompose the mechanism of the complex system of cellular respiration themselves. This relates to literature on student-generated or self-generated models [31,32,51] as opposed to teacher-generated models. Working with self-generated models is preferred as this is done on the basis of students' present skills and knowledge, but research about such models show students to be focused on surface issues rather than on deeper characteristics and systematic processes that develop conceptual understanding [4]. More complex biological topics require a blend of teacher- and self-generated models which rely on already present schema for guidance (in our case Lego® and road plates) and further scaffolding and structuring as well as having the opportunity to recompose and design some of the model segments themselves. In this way, students can re-build the mechanism in order to gain understanding and in such, the modelling process in our research seems to trigger thought processes that follow the ways in which biologists understand complex systems by first decomposing a mechanism into its parts and their operations and then recompose a mechanism based on emergent or decentralized modes of organization on many functional levels [13]. Student questions that relate to such thought processes are for example 'What makes these acids change all the time'? and 'This subprocess only yields two ATP, where is the rest of the ATP produced'? which show that students become more and more focused on higher organizational levels throughout the simulation. However, our research does not provide sufficient data and information to make further claims about these thought processes and more research is needed. In such future research, we think it is important to map the effects on conceptual learning using more instruments that can have a complementary roles in measuring conceptual learning. Also, it should include approaches aimed at measuring the extent to which conceptual learning is transferable to other domains and contexts.

A third issue relating to literature about modelling in simulations and the limitations in our research is that the concrete dynamic model used in our study uses physical, 3D parts which can be related to theories about embodied cognition. Although the importance of embodied cognition is highlighted by many researchers and practitioners, there is no unifying theory of embodiment [52].

As the use of the concrete dynamic model required the use of students' motor systems and they linked their internal imagination to the external model with physical parts, we think that it can be understood as one of the forms of embodied cognition and contribute to the field. For an example of other forms of embodiment for cellular respiration see Ross, Tronson and Ritchie [53]. As we reported great excitement and enjoyment levels visible in smiles, humor and joy entering the classroom, we propose that this form of embodied learning is also important for the affective regulation within complex systems. As some studies have shown, a combination of embodied cognition and computer based models could be a promising direction for future research and development on simulations. As for example Dickes, Sengupta, Farris and Basu [54] have shown, students could then first undergo embodied learning to understand the process and mechanistic patterns, followed by further inquiry and exploration using computer simulations. However, more research on using concrete dynamic models as forms of embodied learning and the combinations with computer simulations is needed [55].

Considering scaffolding and structuring of the modelling process, we can contribute to the notion that offering guidance in the form of structure, instructional support and scaffolding is important [5]. In this study, we had students build up the model gradually around smaller tasks, which connects to earlier work by Lazonder and Kamp [28] who showed that conceptual learning of inquiry tasks in simulations is improved when inquiry tasks are segmented into a series of subtasks. The precise amount and specific forms of scaffolding in our study were designed to represent the ubiquity of microscopic parts and interactions over different organizational levels within the complex system at hand, showing the important role that domain content has on model design. However, as students are known to be very diverse in background, interest and learning styles, we think that differentiation of students into subgroups on the basis of such characteristics and appointing differentiated structuring of the tasks and scaffolding of the model would be an improvement.

In regard to contextualizing the modelling process, we embedded the simulation in an authentic context of a research team wanting to find the efficiency of a sediment battery. We designed this context based on one of the elements described in the Dutch final exam requirements which makes reference to the metabolism of prokaryotes and their applicability in biotechnological context. From literature, we know that finding a context for abstract and complex topics such as cellular respiration can be hard [15] and student interest in cellular respiration in one of the lowest out of the biology curriculum [16,17]. In this research, we have shown that using a biotechnical context in which students become part of a research team wanting to find the efficiency of a sediment battery can lead students to describing the topic as being meaningful to real life. This was also expressed in student questions during the third lesson phase of finding the efficiency of the battery when students become more focused on applicability and usage issues, shown by the questions 'Why don't we use this method for our energy production if we only need 0.4 grams of sugar for an AA battery'? or 'Do all electrons get captured by the graphite, is it 100% efficient'? Considering the use of multiple representation, we first propose that future research should focus more on how all kinds of different forms of dynamic and static models can be combined and how these combinations influence conceptual learning and interest. For example, what would be the effects of using highly interactive paper-pen assignments in comparison to a static model? How do such combinations aid or hinder learning, e.g., how do students recognize where there is a direct similarity between the representation and the target concept and where this breaks down [56]?

*Implications for Practice*

In addition to the theoretical and practical demands that were taken as starting points for the design of a simulation for effective teaching and learning of complex biological systems, this study provides some additional implications for instructors and researchers seeking to design and deploy a similar simulation. First, in designing simulations for complex topics where interest is low, it is important to find and design a concrete dynamic model that makes use of materials that are fun to work with for students (such as Lego®, colored balloons, rubber bands, et cetera) as well as easy to

obtain for teachers. Second, concrete materials and procedures for simulations should be instrumental in and congruent with school settings (e.g., do not use large printed road plates, but rather have teachers print four A3-format papers at school printers and merge them using adhesive tape, or make the procedure easy to follow and prevent unnecessary complexity by providing support right on time). Third, when designing a simulation, provide a correct amount of scaffolds and structure, but also allow a certain degree of student autonomy during the modelling process so that students can make mistakes, for they will learn from their mistakes. Fourth and final, find or design a context in which the bigger idea or rationale behind the entire process of choice is made explicit (in this study being energy transfer from glucose to ATP), for this helps students to understand the ultimate goals of the 'why' of learning complex topics in science education.

In conclusion, this study shows that a simulation using a concrete dynamic model embedded in a context for one of the hardest topics in biology education can result in high situational interest in the form of both excitement and value for real life. Also, this study provides insight into how simulations can influence conceptual learning as well as how to design a dynamic model for complex, abstract topics where students have low levels of prior knowledge and difficulties connecting processes on microscopic levels to more macroscopic, organizational levels.

**Author Contributions:** Conceptualization, all authors; Methodology, M.D., F.J. and C.V.B.; Validation, M.D. and K.O.; Formal Analysis, M.D. and K.O.; Investigation, M.D. and K.O.; Resources, M.D. and K.O.; Data Curation, M.D. and K.O.; Writing M.D.; Writing – Review and Editing, M.D., F.J. and C.V.B.; Visualization, M.D. and K.O.; Supervision, F.J. and C.V.B. All authors read and approved the final manuscript.

**Funding:** This research received no external funding.

**Conflicts of Interest:** The authors declare no conflict of interest.

## Appendix A

Description of the process of cellular respiration.

Cellular respiration is an energy-releasing biological process that consists of a sequence of four sub-processes. The first sub-process is glycolysis, which breaks down a single nutrient molecule, mostly glucose ($C_6H_{12}O_6$) into two molecules of the conjugate base pyruvate ($CH_3COCOO^-$) and also leads to the reduction of two NAD molecules (Nicotinamide Adenine Dinucleotide) into NADH. The second sub process is decarboxylation which removes a carbon atom from the carbon chain in pyruvate, resulting in acetyl-coA. This acetyl-coA then enters the third sub process called the citric acid cycle or Krebs cycle. This cycle starts with a four-carbon acid molecule called oxaloacetate binding to acetyl-coA, forming citric acid. Then, citric acid changes shape through a cascade of molecular changes which leads to more NAD and FAD molecules (flavin adenine dinucleotide) being reduced into NADH ($2H^+ + 4e^- + 2NAD^+ \rightarrow 2NADH$) and $FADH_2$ ($2H^+ + 2e^- + FAD^+ \rightarrow FADH_2$), respectively. In the process, carbon dioxide ($CO_2$) is produced as a waste product. Finally, in the fourth sub process called oxidative phosphorylation or electron transport chain, electrons from electron carriers NADH ($2NADH \rightarrow 2H^+ + 4e^- + 2NAD^+$) and $FADH_2$ ($FADH_2 \rightarrow 2H^+ + 2e^- + FAD^+$) are transferred to the mitochondrial inner membrane by oxidation. Upon receiving energy rich electrons, mitochondria can open selective multiproteins in the membrane that pump protons ($H^+$) outward from the matrix. This gradient of $H^+$ later creates a flux of protons across the membrane down the proton gradient which catalyzes the synthesis of ATP from ADP by ATP synthase. This ATP then serves to transfer and provide energy in organisms. Finally, the now superfluous electrons are accepted by oxygen ($O_2 + 4H^+ + 4e^- \rightarrow 2H_2O$). The electrons thus fulfill a very important function in this biochemical process: they are being transferred from energy rich nutrients like glucose to electron carriers NADH and $FADH_2$ that transfer electrons to the electron transport chain where ATP is formed and electrons are accepted by water that is being excreted.

## Appendix B

The conventional static flow diagrams

**Table A1.** Conventional static flow diagrams as used in this study. © Handbook for the natural sciences and mathematics, BINAS. [45]. 6th ed. Groningen: Noordhoff.

| Sub-process:<br><br>Glycolysis | Subprocesses:<br><br>Decarboxylation<br><br>Citric acid cycle |
| --- | --- |
| 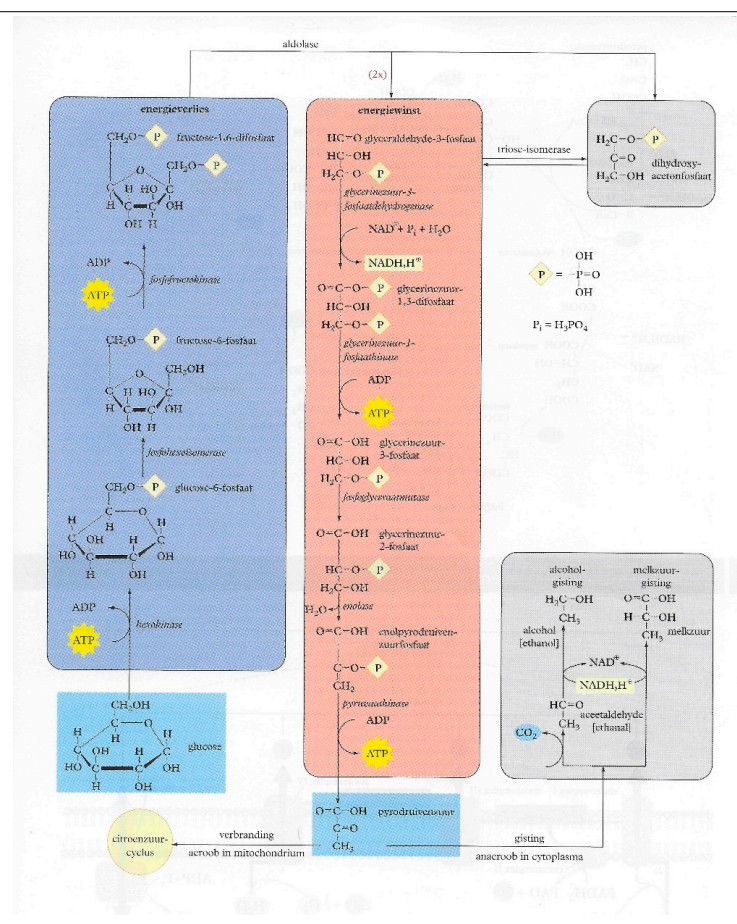 | 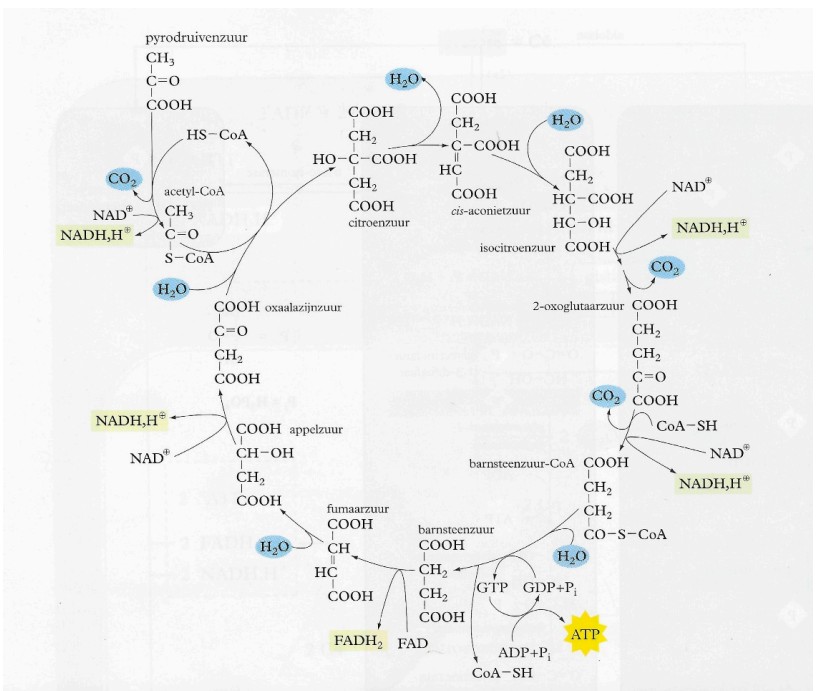 |

## Appendix C

**Table A2.** Survey of the printed Lego®road plates used in this research and impressions of the enactment.

| Glycolysis | Decarboxylation and Citric Acid Cycle |
|---|---|
| 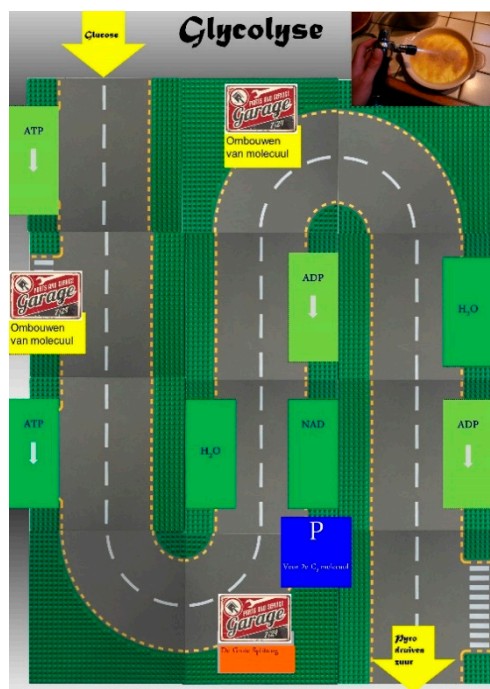 | 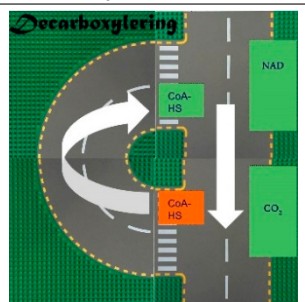<br>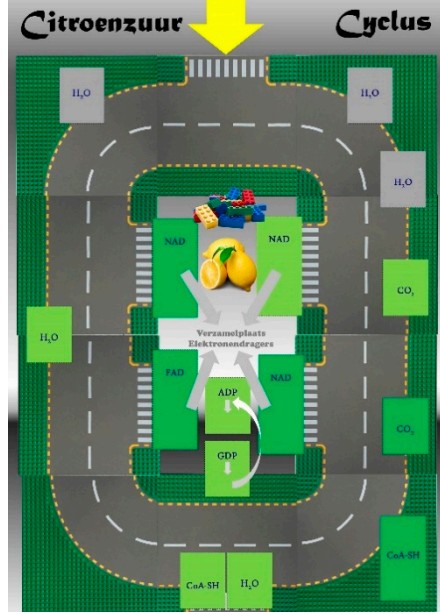 |

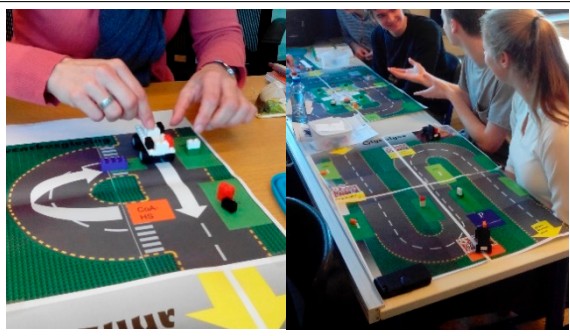

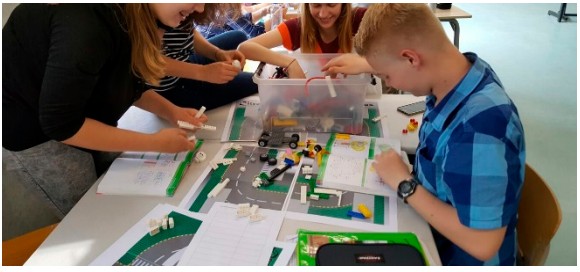

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
