# Peer review of "Understanding Cellular Respiration through Simulation Using Lego® as a Concrete Dynamic Model"

_education, doi:10.3390/educsci9020072_

Round 1
Reviewer 1 Report
In the manuscript “Understanding Cellular Respiration through Simulation 2 using Lego® as a Concrete Dynamic Model”, the authors demonstrate an innovative approach to make students comprehend a complex biological process. The proposed way of teaching this respiration process is shown to be advantageous to conventional teaching.
I think, in general, this study is very relevant to learning as it shows benefits of using simulation of a concrete dynamic model. Even though it is not a novel concept, I have a positive opinion of this manuscript. I have some minor suggestions to further improve the manuscript.
1. While the use of concrete dynamic model has a positive influence on conceptual learning, I believe the limitations in terms of time and accessibility needs to be discussed. More specifically, there should be some discussion about the advantages even if the approach is more time consuming than conventional teaching. Given that, teachers are bound with limited teaching hours most of the time, it is necessary to address the feasibility of having various concrete models. Furthermore, even though Legos are popular and exciting in many countries around the world, the concept is tricky to envision in other countries. Some discussion on other alternatives would make the study more impactful.
2. The study compares the use of dynamic model to the conventional paper-pen assignments. It would be of interest how an in-between approach of an interactive paper-pen assignments with static models would fare. A brief discussion on this would definitely help the readers who want to apply such approach.
3. The metrics used in the study certainly needs to be improved. I would suggest putting the numbers/distributions to graph or plots. While the statistics do not seem to differ much, it would be very helpful to visualize the impact of this approach.
4. The controls are poorly defined in the section “Research Question”. Even though the distinction is on other sections and tables of the manuscript, to help the readers better understand the differences between approaches, I would recommend elaborating on the controls in this section.
5. Please make the citation style in text consistent, for example, references in line 147-148, 515-516.
Author Response
We want to thank you for the review of our manuscript for the special issue Interactive Simulations and Innovative Pedagogy for Conceptual Understanding in Science Education of the journal Education Sciences. In this response letter, we will respond to the comments made by both reviewers and state in what way we have revised the manuscript. The revisions are included in the manuscript by highlighting the changes in bold text marked yellow.

Reviewer 2 Report
This study involves a field experiment comparing the effects of a simulation that is adapted for students to investigate complex biological systems. Outcomes include students' conceptual understanding as well as their situational interest; qualitative data is also analyzed and reported. Overall, this is a timely paper that I think will be of interests to readers of the special issue of Education Sciences. I do, though, have a few concerns about the manuscript as it is written, and so I make some suggestions for revisions to it. I organized my comments into Major and Minor Points.
Major Points
First, I found the connection between the a) need for study, b) research questions, c) methods, and d) results to be unclear. In particular, the research questions focus on students' conceptual understanding, interest development, and the questions student ask. While the need to study students' understanding and interest development was justified well by the literature review, why the questions students ask are in need of study is less clear (and is less justified). Moreover, I was very surprised to see other measures in the Method section - for students' perceptions of the advantages and disadvantages of the simulation and field notes. How these connect to the overall aim of the study is not clear. In addition, how these relate to the research questions is also not clear. I think this is very important to address, as the study is, overall, written with a clear purpose, but these additional measures make it appear like the study is focused (or unclear in its aims/purpose).
Related to the above point, the authors write one research question (What are the effects of simulating cellular respiration within context using Lego® as a concrete dynamic model on conceptual learning and situational interest and what questions do students ask during the learning process?); I think this can be split into three research questions (one each for students' understanding, situational interest, and the questions they ask). Additionally, what question(s) the field notes and students' responses about the advantages and disadvantages answer is not clear.
This is perhaps less important than the first two points, but I suggest the authors include information (in the introduction or literature review or possily the discussion) on how findings from this study inform learning of other challenging complex systems-related topics, i.e., what other phenomena and topics a concrete dynamic model may be useful (in the life sciences or in other scientific disciplines).
Also not as major, but the measures are described in their own sections of the method, whereas I think these would be suited for a Measures or Instruments section of the method section.
Minor Points
Consider emphasizing in the abstract that this study uses an experimental design (is a field experiment).
Consider using language other than “getting a grip” on the phenomenon as a goal for the study - this reads a bit under-specified: does this mean to develop a conceptual understanding of the phenomenon? The capability to investigate it?
The manuscript does a great job of setting up the need to study students’ interest (in light of how disinteresting learning about cellular respiration can be), but not maintained situational interest, particularly. The authors could include a short section to the literature review on interest development (e.g., Hidi & Renninger’s model that Linnenbrink-Garcia and colleagues’ scale is based upon; also, does the 2011 NRC report mention interest?).
At the start of section 4.5, the authors write that they chose two out of the three variables, but it is not immediately clear (though it is from close reading and in the next paragraph( which of the two they chose. Consider clarifying this.
State more about how the means were compared, i.e., for which outcomes? Also, was a dependent or independent samples t-test used (should be dependent, I think).
The authors write that SI-V were both high. While this appears to be the case, for the authors to make it, I suggest carrying out a t-test for each (or, since they are not distinguishable, both) of the means with the null hypothesis being the mid-point of the scale.
What does it mean that five students had an observable ethnic background? What checking was done? Was the teacher/other records consulted? I’d suggest excluding this detail unless it can be supported more carefully.
In the section on participants, research design, and data analysis, the number of students in the treatment and control conditions is only mentioned in the context of the t-test, whereas this is a critical feature of the research design.
I suggest moving data analysis into its own section, and using the first section of the method (perhaps just beneath the method section) to describe the general research design.
For this finding - “No significant 394 main effects were found for the mean of SI-V.”), consider interpreting this slightly more, i.e., state what the absence of a main effect means for the differences between the groups or the effects of the use of the simulation.
Consider italicizing statistics(i.e., Cohen’s d, p-value, t, df, SD).
Perhaps include an implications for practice section offering some ideas for what instructors or researchers seeking to design and employ a similar activity may wish to do.
Author Response

(The authors gave the same response as above.)
